

# Organic Aerosol source apportionment in London 2013 with ME-2: exploring the solution space with annual and seasonal analysis

Ernesto Reyes-Villegas[1], David C. Green[2], Max Priestman[2], Francesco Canonaco[3], Hugh Coe[1], André S.H. Prévôt[3], James D. Allan[1,4]

[1]School of Earth, Atmospheric and Environmental Sciences, The University of Manchester, Manchester, M13 9PL, UK
[2] School of Biomedical and Health Sciences, King's College London, London, UK
[3] Laboratory of Atmospheric Chemistry, Paul Scherrer Institute, 5232 Villigen PSI, Switzerland
[4] National Centre for Atmospheric Science, The University of Manchester, Manchester, M13 9PL, UK

*Correspondence to*: James D. Allan (james.allan@manchester.ac.uk)

**Abstract.** The Multilinear Engine (ME-2) factorisation tool is being widely used following the recent development of the Source Finder (SoFi) interphase at PSI. However, the success of this tool, when using the a-value approach, largely depends on the inputs (i.e. target profiles) applied as well as the experience of the user. A strategy to explore the solution space is proposed to objectively determine the solution that best deconvolves the organic aerosol (OA) sources where trilinear regression has proven to be a useful tool to compare different ME-2 solutions. Aerosol Chemical Speciation Monitor

(ACSM) measurements were carried out at the urban-background site of North Kensington, London from March to December 2013, where for the first time the behaviour of OA sources and their possible environmental implications are studied using an ACSM. Five OA sources were identified: biomass burning OA (BBOA), hydrocarbon-like OA (HOA), cooking OA (COA), semivolatile oxygenated OA (SVOOA) and low-volatility oxygenated OA (LVOOA). ME-2 analysis of the seasonal datasets (spring, summer and autumn) showed a higher seasonal variability in the OA sources that was not

detected when the March-December dataset was analysed; this variability was explored with the triangle plots f44:f43 f44:f60, with HOA and COA being the most suitable sources to constrain. Further analysis on the atmospheric implications of these OA sources was carried out, identifying evidence of the possible contribution of heavy-duty diesel vehicles to air pollution during weekdays compared to those fuelled by petrol.

Keywords: Aerosol sources, matrix factorization, air quality, $PM_1$, ACSM, SoFi.

**1. Introduction**

Developed countries have made great improvements in air quality. However, air pollution still represents a significant air quality issue, mainly in urban cities, due to the sheer number of inhabitants and the associated anthropogenic emissions resulting from the inhabitants' daily activities (transportation, energy production and industrial activities). In particular, aerosols have significant effects on air quality (Watson, 2002;Pope III and Dockery, 2006;Keywood et al., 2015).

Organic aerosols (OA) are one of the main constituents of submicron particulate matter, comprising 20–90% of the total submicron particle mass (Zhang et al., 2007). OA are classified according to their origin as primary OA (POA) or secondary OA (SOA). POA are directly emitted from a range of sources while SOA are produced from gaseous precursors (VOCs) by chemical reactions in the atmosphere. POA sources range from traffic emissions (hydrocarbon-like OA, HOA), biomass burning OA (BBOA), to OA emissions from cooking (COA) among others. Kupiainen and Klimont (2007) determined that

the main sources of POA in Europe were emissions from traffic and the residential combustion of solid fuels. Allan et al. (2010) identified three POA sources: transport, burning of solid fuels and cooking in Manchester and London. SOA are the main constituents of OA ranging from 64% in urban areas to 95% in rural sites (Zhang et al., 2007). Previous source apportionment studies (Zhang et al., 2011) often identified a highly oxygenated fraction with low volatility (LVOOA) and a



less oxygenated and more volatile species (SVOOA). In general, SVOOA represent fresh SOA, which, after photochemical

processing, evolve into LVOOA (Jimenez et al., 2009). POA and SOA concentrations vary over seasons and years, thus in order to study the OA sources and processes as well as their impacts on air quality, it is necessary to carry out long-term measurements and subsequent source apportionment data analysis.

Aerosol mass spectrometry has been widely used for measuring aerosol concentrations in a wide range of ground based measurements (Hildebrandt et al., 2011;Mohr et al., 2012;Saarikoski et al., 2012;Young et al., 2015b). In particular, the

Aerosol Chemical Speciation Monitor (ACSM), which has been recently developed (Ng et al., 2011), has been used to carry out long-term measurements of non-refractory submicron aerosols around the world, for instance an industrial-residential area in Atlanta, Georgia (Budisulistiorini et al., 2013), at background locations in South Africa, (Vakkari et al., 2014) and Spain (Minguillón et al., 2015a;Ripoll et al., 2015), a semi-rural site in Paris (Petit et al., 2015) and at an urban background site in Switzerland (Canonaco et al., 2015).

Source apportionment techniques have been widely used to quantitatively determine aerosol sources. The main source apportionment models include: chemical mass balance (CMB) and positive matrix factorization (PMF).

CMB uses prior knowledge of source profiles and assuming that the composition of all sources is well defined and known (Henry et al., 1984). This technique is ideal when changes between the source and the receptor are minimal, although this barely happens in real atmospheric conditions and the constraints may add a high level of uncertainty.

PMF is a least-squares approach based on a receptor-only multivariate factor analytic model  (Paatero and Tapper, 1994) . The main difference between PMF and CMB is that PMF does not require any information as input to the model and the profiles and contributions are uniquely modelled by the solver (Paatero et al., 2002). PMF was applied to OA data measured with an AMS for the first time by Lanz et al. (2007), using measurements taken at an urban background site in Zurich in the summer of 2005, where six OA sources were determined: LVOOA, SVOOA, HOA, Charbroiling-like OA, BBOA and COA.

Subsequently, PMF was successfully applied to other datasets, acquired from a wide range of sampling sites and with different techniques, Ng et al. (2010) compiled and analysed 43 studies carried out at different sites around the world. This study provided a broad overview of aerosol composition and the importance of SOA as well as BBOA and HOA sources. In other PMF studies, it was possible to find other relevant sources such as COA (Allan et al., 2010;Huang et al., 2010;Liu et al., 2012;Mohr et al., 2012;Sun et al., 2013;Crippa et al., 2013a).

ME-2 is a multivariate solver that determines solutions using the same equations as PMF (Paatero, 1999), with the possibility of using previous knowledge (factor time series and / or factor profiles) as inputs to the model, to partially constrain the solution, thereby reducing the rotational ambiguity (Paatero et al., 2002) and leading to more interpretable PMF solution(s) as shown in Lanz et al. (2008) where three sources of OA were successfully determined (traffic related, solid fuel and secondary OA) during winter in an urban-background site in Zurich. Here unconstrained PMF runs failed to identify the

environmental solution. This was most probably due to a high degree of temporal co-variation in the OA sources driven by low temperatures and periods of strong inversion.

The development of the Source Finder (SoFi) interphase (Canonaco et al., 2013) written on the software package Igor Pro (WaveMetrics, Inc.) together with a further standardised approach developed by Crippa et al. (2014), allowed different OA source apportionment studies to be undertaken such as a study at a suburban background site in Paris, France during January-

March 2012 (Petit et al., 2014); laboratory studies analysing atmospheric ageing from the photo-oxidation of α-pinene and of wood combustion emissions in smog chambers and flow reactors (Bruns et al., 2015); long-term measurements (February 2011-February 2012) carried out at an urban background site in Zurich, Switzerland studying differences in oxygenated OA during summer and winter periods (Canonaco et al., 2015). As part of the ACTRIS project (Aerosols, Clouds, and Trace





gases Research InfraStructure network) (Fröhlich et al., 2015) an intercomparison between 14 ACSM and one high
resolution time of flight aerosol mass spectrometer (HR-ToF-AMS) was carried out at the SIRTA site in Gif-sur-Yvette near
Paris, being able to identify 4 sources: hydrocarbon-like OA (HOA), OA related to cooking activities (COA), biomass
burning related OA (BBOA) and oxygenated organic aerosol (OOA). These four sources were successfully identified from
HR-ToF-AMS measurements with unconstrained PMF analysis. However, in the case of the ACSM datasets, it was
necessary to partially constrain solutions via ME-2 analysis; probably due to the low signal to noise ratio of ACSM data

compared to the AMS and the rural site type. Furthermore, new ME-2 source apportionment studies have been published this
year (Bozzetti et al., 2016;Fountoukis et al., 2016;Milic et al., 2016;Elser et al., 2016), and even more are expected to come
due to the successful application of SoFi. Thus, new strategies to objectively explore the solutions are needed.

This study includes data analysis of the first ACSM instrument deployed in the UK at the North Kensington site from March
to December 2013, using the recently developed graphical interphase SoFi, to perform non-refractory OA source

apportionment analysis with the ME-2 factorization tool, implementing a strategy to determine the solution that best
identifies OA sources and with further subsequent discussion of the various identified OA sources.

## 2. Methodology

The data used in this analysis (5th March – 30th December 2013) were obtained using an Aerosol Chemical Speciation
Monitor (ACSM), deployed at the urban-background site in North Kensington, London. This instrument is owned by The

Department for Environment, Food and Rural Affairs (DEFRA) and is part of the Aerosols, Clouds, and Trace gases
Research InfraStructure Network (ACTRIS).

Source apportionment of OA was carried out using the PMF model implemented through the Multi-linear Engine tool (ME-
2) and controlled via the Source Finder (SoFi) graphical user interphase version 4.8, developed at the Paul Scherrer Institute
(PSI), Switzerland (Canonaco et al., 2013).

### 2.1 Site and instrumentation

North Kensington (51.5215°, -0.2129°) is an urban background site located adjacent to a school, 7 Km to the west of central
London. There is a residential road 30 metres to the east with an average traffic flow of 8,000 vehicles per day (Bigi and
Harrison, 2010). This monitoring site is part of the DEFRA Automatic Urban and Rural Network ([http://uk-air.defra.gov.uk/networks/network-info?view=aurn](http://uk-air.defra.gov.uk/networks/network-info?view=aurn)).

As an urban background site, North Kensington is not significantly influenced by a single source or street, and
concentrations may be analysed as an integrated contribution from all sources upwind of the site in London. This site is
widely accepted as representative site of background air quality in central London and has a large set of long-term
measurements for various pollutants (Bigi and Harrison, 2010). Different studies have been carried out at this site such as
analysis of elemental and organic carbon concentrations in offline measurements of particulate matter with a diameter less

than 10 micrometres ($PM_{10}$) (Jones and Harrison, 2005), aerosols and $NO_x$ association with wind speed (Jones et al., 2010),
properties of nanoparticles (Dall'Osto et al., 2011), coarse particles (Liu and Harrison, 2011) and in aerosol chemical
composition (Beccaceci et al., 2015) in the atmosphere. The first long-term study of the behaviour of non-refractory
inorganic and organic aerosols at the North Kensington site was carried out analysing cToF-AMS data collected from
January 2012 to January 2013 (Young et al., 2015a) where source apportionment analysis was carried out applying

unconstrained PMF runs, with five sources identified: HOA, COA, solid fuel OA (SFOA), SVOOA and LVOOA.




The Aerosol Chemical Speciation Monitor (ACSM) measures, in real time, the mass and chemical composition of particulate organics, nitrate ($NO_3$), sulphate ($SO_4$) , ammonium ($NH_4$) and chloride (Cl) ions, with a detection limit of 0.2 µgm$^{-3}$ for an average sampling time of 30 min (Ng et al., 2011). These chemical species measured by the ACSM are determined according to the same methodology used in the AMS as defined by Allan et al. (2004). In principle, the ACSM is designed and built under the same sampling and detection technology as state-of-the-art Aerosol Mass Spectrometer (AMS) instruments. However, the ACSM is better suited for air quality monitoring applications due to its lower size, weight, cost, and power requirements; it is also more affordable to operate and is capable of measuring over long periods of time without supervision (Ng et al., 2011).

Time series of pollutants such as BC, CO, $NO_x$, OC, EC were downloaded from the DEFRA website for the North Kensington monitoring site. Wind speed and direction data were obtained from the meteorological station at Heathrow airport (located 17 Km from the sampling site). Wind data from this site was used due to its representativeness of regional winds without being affected by surrounding buildings.

**2.2 Source apportionment (ME-2)**

The multilinear engine algorithm (Paatero, 1999) is a multivariate solver that is typically used to solve the PMF model, which is based on a receptor-only factor analytic model (Paatero and Tapper, 1994). The bilinear representation of PMF solves Eq. (1), written in matrix notation, which represents the mass balance between the factor profiles and the concentrations.

$$X = G * F + E \qquad (1)$$

The elements $g_{ik}$ of matrix G represent the time series and the elements $f_{kj}$ of matrix F represent the j elements of the profile (for example, mass spectrum). E is the model residual

The parameters $f$ and $g$ are fitted using a least squares approach that iteratively minimizes the variable Q (Paatero et al., 2002).

$$Q(f,g) = \sum_{i=1}^{m}\sum_{j=1}^{n}\left(\frac{e_{ij}}{\sigma_{ij}}\right)^2 \qquad (2)$$

Where, $e_{ij}$ represent the residuals and $\sigma_{ij}$ the estimated uncertainty for the points $i$ and $j$.

The variable Q depends on the number of selected factors and the size of the data matrix, hence it is necessary to normalize Q by the degree of freedom of the model solution ($Q_{exp}$) (Paatero et al., 2002) to monitor solutions.

$$Q_{exp} \cong n * m - p * (m + n) \qquad (3)$$

Where $p$ is the number of factors chosen, $n$ number of samples and $m$ the mass spectra. Ideally, if the model accurately captured the variability of the measured data, it would be expected to have a value of $Q/Q_{exp} = 1$, still this value depends on fluctuations in the source profiles, over- or underestimation of input data errors and from the model error.

Solutions using a least squares approach to solve a factor analysis problem may have linear transformations, also known as rotations (Paatero and Hopke, 2009). One advantage of ME2 over PMF is that the rotational ambiguity can be reduced by using previous knowledge of profiles (for example mass spectra) or time series of different pollutants using e.g. the $a$-value approach. The $a$-value is a parameter that represents the degree of variability of the target profile, which typically ranges from zero to one, the closer to zero the more constrained the solution is (Lanz et al., 2008). The user should keep in mind





that partially constrained solutions are carried out by compromising the $Q/Q_{exp}$ value, which should be monitored to determine the feasibility of the solutions.

### 2.2.1 Target profiles and levels of constraint

In this study, solutions obtained with ME-2 were constrained using the *a*-value approach, by using four different sets of mass spectra from previous studies as target profiles (Table 1). The "a" set of target profiles represent BBOA and HOA average factor profiles obtained from an analysis carried out on different mass spectra from a variety of monitoring sites across Europe (Crippa et al., 2014) and COA obtained from a study in Paris (Crippa et al., 2013a). "c", "s" and "w" sets of target profiles (TP) were provided by Young et al. (2015a) from a PMF analysis carried out on AMS measurements at the North Kensington site in London, 2012. "c" TP were obtained from an analysis performed on annual OA measured with a cToF-AMS (11 January 2012 – 23 January 2013). "s" and "w" TP were obtained from summer and winter measurements were taken with an HR-AMS (January–February 2012 and July–August 2012, respectively).

A wide range of combinations of TP and *a*-values were used during this analysis, all of them being run with three random initial values (seeds), to determine the stability of the solutions. Constraints were applied using one, two and three TP; in all the solutions, there were at least two unconstrained factors. Figure 1 shows the coding used to identify the different solutions, for example when constraining three factor profiles: wB5_H2_C3_S1.

### 2.3 Strategy to explore the solution space

The success of ME-2 relies on the additional use of *a priori* information in form of constraints. However, without a well-defined strategy or a limited analysis of the solution space, it may lead to a subjectively and inaccurately selected solution. Moreover, where possible, target profiles from different studies should be tested in order to determine which set of target profiles are the most adequate. Therefore, the following sections show the results of the analysis carried out on the dataset March-December 2013, where the considerations provided by Crippa et al. (2014) were applied. Moreover, new analysis techniques were developed to explore the solution space.

PMF solutions are run to determine the number of factors (sources) in the solution; this is carried out by running PMF for a different number of factors. Once the number of possible sources has been chosen, different combinations of *a*-values and constrained factors are tested to determine the solution that better identifies the OA sources. The residual of the solution provides important information; it is possible to determine if the solution is over estimated (negative residual) or under estimated (positive residual). When a structure on the diurnal residual is observed, it allows the factor which is affecting the residual to be determined (Crippa et al., 2014) and a decision to be made if the *a*-value should be modified or even if the target profile is the appropriate or not for this dataset. Together with the residual, it is recommended to look at the total $Q/Q_{exp}$, which is a parameter used to monitor solutions, the best solution will be the one with values closest to one.

Trilinear regression is used as a new technique to explore the solution space in ME-2 analysis. Multilinear regression has been previously applied to analyse the relationship between POA and combustion tracers (Allan et al., 2010;Liu et al., 2011;Young et al., 2015b) as well as polycyclic aromatic hydrocarbons (Elser et al., 2016). This is used instead of simple linear regression because many of the combustion related variables will have multiple sources such as biomass burning and traffic. Eq. (4) shows the trilinear regression equation used to analyse the relationship between POA and combustion tracers.

$$Y = A + B[BBOA] + C[HOA] + D[COA] \tag{4}$$

Where "Y" is $NO_x$, BC, or CO.





The following considerations should be taken into account: The slopes and intercepts should be positive as they represent air pollutant concentrations (the intercept A is representative of the 'background' concentration); the slope D for COA is used as a validation parameter which should be close to zero, due to its low contribution to BC, $NO_x$ and CO, owing to the fact that most cooking in the UK uses electricity or natural gas as a source of heat. A nonzero value would indicate correlation with combustion tracers and thus the possibility that it is receiving interference from HOA, which has a similar mass spectrum.

Chi-square is used as a "goodness of fit" where the lower the value the better fit between the analysed pollutants.

### 3. Results

### 3.1 Exploring the solution space for March-December dataset

This section shows the results from the analysis applied to determine the solution that best represents the OA sources for the complete dataset March-December 2013, where a total of 25 solutions were analysed.

### 3.1.1 Solutions, *a*-values and stability

By analysing unconstrained runs, it was possible to determine a solution with five factors (Fig. S1.b, BBOA, HOA, COA, SVOOA, LVOOA). With the number of factors determined, ME-2 is run using a range of *a*-values. These *a*-values were selected after trial and error and according to the literature (Lanz et al., 2008;Crippa et al., 2014;Petit et al., 2014), which suggests that *a*-values depend on the similarity of the target profile and the factor profile being analysed: HOA mass spectra

do not show high variability when compared with different sites, thus it is possible to restrict the constraint with *a*-values of 0.1-0.2. On the other hand, COA and BBOA mass spectra from different sites show high variability and a looser constraint should be applied (for example *a*-values 0.3-0.5 or higher).

Constraining only one or two factors of the five-factor solutions gave the least favourable results with high residuals and mixing factor profiles. When analysing the different seeds, these solutions also showed high variability between seeds;

greater stability was found when three of the five factor solutions were constrained (Fig. S2), behaviour also observed by Crippa et al. (2014). As a result, in this analysis, five-factor solutions constraining three factors will be analysed for the first seed. One PMF solution and two solutions constraining two factors were also used during the exploration (Fig. 2) for three sets of TP.

### 3.1.2 Q/Q$_{exp}$, diurnal residual and trilinear regression

Currently, there is not a standard criterion to define a satisfactory Q/Q$_{exp}$ value as a certain amount of 'model error' will cause it to be systematically higher than unity (Ulbrich et al., 2009).When analysing different solutions from the same dataset (Fig. 2.b), it is possible to observe that the use of different *a*-values does not imply a high variation, ranging between 1.88-2.2, suggesting that all the solutions are mathematically acceptable. The unconstrained solution is the one with the lowest total Q/Q$_{exp}$ with a value of 1.88, which is expected, as PMF calculates the solution by minimising this value;

however, PMF solution has a high Chi square and negative slope for COA (Fig. 2.a), suggesting that this solution is not environmentally acceptable, thus it is necessary to analyse all the different parameters in fig. 2 in order to select the solution that best identifies the OA sources.

Figure 2.a shows the diurnal residual analysis where solutions constrained with "c" target profiles present high positive residual around 14:00-19:00 hrs. Solutions constrained with "w" target profiles have a negative residual during early

morning with a positive residual at 21:00 hrs. Hence, the solution with a better diurnal residual is within the solutions constrained with "a" target profiles.



Figure 2.b shows the trilinear regression outputs between $NO_x$ and POA for the different solutions (see supplement S3 for BC and CO trilinear regressions). All the solutions properly identified the background $NO_x$ concentrations (grey line). Solutions with "c" and "w" target profiles showed similar undesirable results as in the diurnal residual analysis, with "c"

target profiles presenting negative COA slopes and "w" presenting high COA slopes and Chi square values, consistent with the diurnal residual analysis that the best solution is with the solutions constrained with "a" target profiles. Additionally, trilinear regression outputs show variations between different solutions constrained with "a" target profiles with changes mainly in the Chi square and the BBOA

### 3.1.3 Diurnal concentrations and mass spectra

OA sources have characteristic diurnal trends, and they may be used, together with their respective mass spectra, to analyse the solutions and determine if all the factors in the solution are suitable, environmentally speaking. BBOA showed low concentrations during the day, with high concentrations at night, mainly related to domestic heating (Alfarra et al., 2007); HOA presents two peaks during the day related to commuting, one in the morning and another one in the evening (Zhang et al., 2005); COA has two peaks related to OA emissions from cooking activities: one peak at noon and one peak in the

evening (Allan et al., 2010). SVOOA is temperature dependent with low concentrations during the day increasing in the evening due to the condensation of gas phase pollutants. LVOOA, due to its regional origin, does not show high variations in its diurnal trend.

Diurnal concentrations for all the solutions (Supplement S3) were analysed to determine the main sources. Here, it was possible to observe that solutions with undesirable outputs in the residual, total $Q/Q_{exp}$ and/or trilinear regression were likely

to have mixed diurnal concentrations between two sources, for example, in the case of HOA, with high concentrations during the evening (Fig. S4.c.16).

These undesirable outputs previously observed were also detected when analysing the mass spectra of the different solutions; Fig. S3 shows examples of diverse situations that were found: in the solution wB7_H5_C7_S1 it is possible to observe mixed factors where SVOOA has peaks of BBOA (m/z 60) and COA (m/z 55 and 57); one source with only one strong peak

in its mass spectrum (SVOOA in solution cB3_H1_S1); The PMF solution was not able to properly identify a BBOA factor with low peaks at m/z 60 and 73.

Finally, from this analysis, aB3_H2_C3_S1 was determined to be the solution that best represents the OA sources for March-December analysis.

### 3.2 Seasonal analysis

When applying source apportionment, ME-2 considers that both target profiles and factor profiles remain constant over time, which may not be the case for long periods of time where meteorological conditions and pollutant emissions related to human activities vary greatly (Canonaco et al., 2015;Ripoll et al., 2015), thus the same analysis that was carried out on March-December data set was applied to data divided into seasons of the year: spring (March, April and May), summer (June, July and August) and autumn (September, October and November), see supplement S.3 for detailed information of the

seasonal analysis.

Analysing the spring dataset (Fig. S5), solutions constrained with "a" and "c" TP were found to present the least favourable results with high Chi-square values and negative COA ratios in the trilinear analysis, as well as a higher negative diurnal residual; the solution wB3_H1_C3_S1 was deemed to be the best solution for spring analysis. Solutions constrained with "s" and "c" TP were the least favourable results for the summer analysis (Fig. S6) with low chi-square values in "s" target profiles, which show high negative residuals in the morning and at night. "c" target profiles show a high positive residual





around 15:00-18:00 hrs; the solution aB5_H1_C3_S1 was found to be the best solution for the summer analysis. In the autumn analysis (Fig. S7), solutions constrained with "a" and "w" TP were found to be the least favourable results with high positive residuals in the morning, also "a" target profiles show high chi-square values. The solution cB3_H1_S1 was deemed the best solution for the autumn analysis.

## 4. Discussion and atmospheric implications

### 4.1 Annual and seasonal solutions

In the following subsections, the outputs of annual and seasonal solutions are compared in order to further explore the variability of the different OA sources.

### 4.1.1 Total $Q/Q_{exp}$ and diurnal residual

Analysing the total $Q/Q_{exp}$, all the solutions obtained were mathematically acceptable and with small variations between their different values: 1.95 for March-December, 2.01 for spring, 1.95 for summer and 1.96 for autumn (Fig. 3.a).

$Q/Q_{exp}$ values obtained in this study are compared to values obtained in different ME-2 studies, for example Petit et al. (2014), in a study using an ACSM, obtained a $Q/Q_{exp}$ value of 6; studies carried out in Spain during winter and summer obtained 1.15 and 0.38 respectively (Minguillón et al., 2015b). $Q/Q_{exp}$ values obtained with PMF are also comparable with values obtained in this study, for example (Young et al., 2015a) obtained a value of 1.35 from annual measurements carried out with a cToF-AMS at this site; (Allan et al., 2010) obtained different $Q/Q_{exp}$ values for the analysis carried out to three different data sets: a value of 3.9 from measurements obtained using a HR-ToF-AMS and values of 10.5 and 16.7 using a cToF-AMS; also Crippa et al. (2013b) identified a $Q/Q_{exp}$ value of 4.59, on HR-ToF-AMS measurements during July 2009 at the urban background site in Paris. Due to all this variability of $Q/Q_{exp}$ values found in the literature, this parameter alone cannot be used as a criterion to determine the solution that best identifies the OA sources.

It is in the diurnal residual where we can observe a high variation (Fig. 3.b), with autumn proving to be the most overestimated with negative residuals of -0.033 mainly in the morning and at night. On the other hand, summer shows to be the most underestimated solution with values of 0.018 particularly between midday and 17:00 UTC. The fact that summer is overestimated from 12:00 to 17:00 UTC is probably related to the increase on photochemical activity and hence a strong variation of the source profiles, a situation that ME-2 is not able to capture.

### 4.1.2 Trilinear regression analysis

Figure 3.a shows that $NO_x/HOA$ ratios present high variability with values of 50.0 for March-Dec, 81.0 for spring, 41.0 for summer and 85.5 for autumn. Comparing $NO_x/BBOA$ and $NO_x/HOA$, a higher ratio is expected for $NO_x/HOA$ due to HOA and $NO_x$ being traffic related pollutants. In Fig. 3.a it is possible to observe that seasonal solutions present high variation for HOA and BBOA compared to March-Dec solution, which suggests that there are seasonal variations with that the March-Dec dataset solution does not completely capture. With regard to background and COA concentrations, they are well identified and relatively constant over the different solutions.

All this analysis carried out in the section 4 shows that seasonal analysis more accurately deconvolves OA sources, being possible to obtain more detailed information that will be lost when running ME-2 for long periods of time.





### 4.1.3 Target profiles (TP) and their impact on the solutions

As previously mentioned, the best solutions were: aB3_H2_C3_S1 for March-December, wB3_H1_C3_S1 for spring, aB5_H1_C3_S1 for summer and wB3_H1_S1 for autumn. The fact that March-December and summer solutions were obtained with "a" TP is possibly due to the fact that these TP represent an average from different mass spectra, becoming robust TP able to deal with the variations of these two datasets; one large dataset (March-Dec) and one dataset with concentrations affected by the different photochemical processes due to the high temperatures (Summer). On the other hand, spring and autumn do not show these variations and their OA sources may be apportioned using winter TP which were obtained under similar temperatures.

Looking at the "c" and "s" TP, these were the ones with the least favourable results in all analyses carried out. This may be attributed to "c" being the only TP obtained with a cToF-AMS while the rest were obtained using a HR-AMS. In the case of "s" TP the unfavourable outputs are again related to the high variability present during this period of time. This analysis shows the importance of using the adequate TP when doing source apportionment as well as to explore solutions with different types of TP in order to objectively determine the OA sources.

### 4.2 Variability of factor profiles

The variability of the different solutions previously obtained may be explored further with the triangle plots f44 vs f43 (Ng et al., 2010;Morgan et al., 2010) and f44 vs f60 (Cubison et al., 2011). The parameters f43, f44 and f60 represent the ratio of the integrated signal at m/z 43, 44 and 60, respectively, to the total signal in the organic component mass spectrum. Figure 4.a shows that BBOA, HOA, COA and LVOOA, while having different values between solutions, are found in distinct areas of the plot, whereas SVOOA shows values of 44:43 with high variability. This variability is displayed with the coloured lines connecting SVOOA and LVOOA of their respective solutions (Fig. 4.a). This analysis shows that the factors derived for SOA do not always conform to the model of LVOOA and SVOOA proposed by Jimenez et al. (2009). Furthermore, the fact that the lines are going in different directions with the seasons of year means that the factorisation is identifying different aspects of the chemical complexity. This serves to highlight that a 2-component model (LVOOA and SVOOA) is an oversimplification of a complex chemical system and further work may be required to completely explain SOA.

By analysing Fig. 4.b, it is possible to observe the variability in f60, with the lowest value obtained in summer (0.013) followed by spring, autumn and March-Dec (0.022, 0.024 and 0.034, respectively). Variability in biomass burning OA depends on the fuel type, burning conditions and level of processing (Weimer et al., 2008;Hennigan et al., 2011;Ortega et al., 2013;Young et al., 2015b). A study carried out by (Young et al., 2015b) in London, 2012, identified two types of solid fuel OA factors, attributed to differences in burning efficiency. Chemical ageing of BBOA has been frequently observed with high f44 values and low f60 due to photochemical processing (Huffman et al., 2009;Cubison et al., 2011). Thus, it was possible to obtain a variety of SVOOA and BBOA mass spectra for the different seasons of the year, ranging from a fresh OA during autumn to a highly oxygenated BBOA during summer.

For all the solutions, COA presents an f60 value of approximately 0.01, which has been previously identified by Mohr et al. (2009) who obtained f60 values of 0.015 - 0.03 for different types of meat cooking. The fact that all the COA mass spectra present similar f44:f60 ratios suggest that COA footprint is relatively constant over the different seasons, being together with HOA the more adequate sources to constrain when applying the *a*-value approach.

### 4.3 Petrol and diesel contribution to traffic emissions

Traffic emissions contribute significantly to air pollution (Beevers et al., 2012;Carslaw et al., 2013;May et al., 2014). In order to better analyse traffic emissions and their impact on air quality, it is necessary to understand the fuel type and





pollutant contribution from different vehicles. In particular, the United Kingdom has a considerable percentage of diesel-fuelled vehicles; according to the vehicle licencing statistics, the percentage of diesel-fuelled vehicles licenced has been increasing over the last few years from 22% in 2006 to 36.2% in 2014 respectively, while petrol-fuelled vehicles decreased from 77.7% to 62.9% (GOV.UK, 2015).

Diesel emits higher $NO_x$ and HOA concentrations compared to petrol, while petrol emits higher concentrations of CO. Moreover, there are variations between Light Duty Diesel (LDD) and Heavy Duty Diesel (HDD) emissions, with LDD emitting higher $NO_x$ concentrations and HDD emitting higher HOA concentrations.

One way to analyse the impact of different fuels on air pollution is to determine weekday/weekend ratios (WD/WE) (Bahreini et al., 2012;Tao and Harley, 2014;DeWitt et al., 2015). This analysis is performed considering WD as Monday to Friday and WE only Sunday to eliminate the mixed traffic on Saturday. Another consideration is that heavy duty/light duty emissions fleet ratio is higher during the week (Lough et al., 2006;Bahreini et al., 2012;Heo et al., 2015). It is also important to state that heavy duty vehicles are exclusively diesel fuelled whereas light duty vehicles are a mixture of diesel and petrol.

Trilinear regression explained in section 2.3, was used with data divided into WD (Monday to Friday) and WE (Sunday) to analyse the WD/WE contributions. Subsequently, it was possible to determine WD/WE ratios for the slopes: $NO_x$/HOA and CO/HOA.

In order to compare these trilinear outputs with the WD/WE ratios between $NO_x$ and CO, $NO_x/\Delta CO$ was calculated from average concentrations: there is a difference in lifetime between CO (lifetime of months) and $NO_x$ (lifetime of hours), thus it is important to consider the background CO concentrations to be able to compare $NO_x$ and CO concentrations. It is necessary to perform a linear regression between CO and $NO_x$ and calculate $\Delta CO$, which is the average CO concentration minus the intercept from the CO:$NO_x$ linear regression.

Figure 5 shows the WD/WE ratios, where it is possible to observe $NO_x/\Delta CO$ ratios of 1.25, 1.35 and 1.136 for March-Dec, summer and autumn, respectively, suggesting diesel with a higher contribution during WD compared to petrol. These findings are confirmed by the CO/HOA ratios, which for the same periods of time, are lower than one (0.8, 0.45 and 0.9) suggesting a lower contribution of petrol during weekdays compared to diesel. In Spring, there are no considerable changes to the WD/WE ratios, although it is possible to observe a higher contribution of petrol during WD with values of 1.28 for CO/HOA and low diesel contribution. Analysing the $NO_x$/HOA ratios, the seasonal ratios show values of 1.07, 1.06 and 1.05 suggesting a slightly higher contribution of LDD during WD than HDD.

### 4.4 PM$_{2.5}$ daily concentrations and PM$_1$ composition

PM$_{2.5}$ has been widely studied due to its potential to cause negative effects on health (Pope III and Dockery, 2006;Harrison et al., 2012;Bohnenstengel et al., 2014). This adverse impact is directly connected to the size of the particles, making PM$_1$ more detrimental to health than PM$_{2.5}$ (Ramgolam et al., 2009). Moreover, analysing aerosol contribution to PM$_1$ and its association with PM$_{2.5}$ concentrations allows the possible influence of PM$_1$ on PM$_{2.5}$ levels to be determined. According to the Daily Air Quality Index (DAQI), PM$_{2.5}$ concentrations are considered moderate when daily concentrations are between 35 and 52 $\mu gm^{-3}$ and high when levels are between 53 and 69 $\mu gm^{-3}$. Daily PM$_{2.5}$ concentrations during the sampling period show that the majority of daily concentrations were considered to be low episodes (Fig. 6), with 10 episodes of moderate concentrations and only two episodes of high PM$_{2.5}$ concentrations (55.2 and 61.5 $\mu gm^{-3}$).

The main PM$_1$ contributors to moderate and high PM$_{2.5}$ concentrations are $NO_3$ and LVOOA. All these episodes were observed with low wind speeds (Fig. S8); however, different contributions from OA sources were identified: in the episode



in March, high BBOA concentrations were observed, whereas during the episodes in April and September, high concentrations of LVOOA were detected .

As a result, the main $PM_1$ contributors to $PM_{2.5}$ concentrations are secondary aerosols. These findings agree with a previous study carried out in this same monitoring site carried out by (Young et al., 2015a) who found secondary aerosols to dominate high concentrations over the year, with different secondary inorganic and organic aerosol contributions between winter and summer.

**5. Conclusions**

This study presents the source apportionment carried out using ME-2 within SoFi 4.8 of OA concentrations, measured with an ACSM from March to December 2013 at the urban-background site in North Kensington, London; the first time it was deployed in the UK.

ME-2 proved to be a robust tool to deconvolve OA sources. This study highlighted the importance of using appropriate mass spectra as target profiles and *a*-values when exploring the solution space. With the implementation of new techniques to compare different solutions, it was possible to show their variability and to objectively determine the best solution, mathematically and environmentally speaking. The comparison carried out between the solution for the March-December dataset and the seasonal solutions showed high variations mainly in the SVOOA mass spectra and the BBOA; less variability was observed in LVOOA, COA and HOA. These variations infer the importance of running ME-2 during periods of time where weather conditions and emissions from human activities are less variable, such as seasonal analyses.

SVOOA presented a high variability in the oxidation state during the different seasons. This is due to the nature of SVOOA showing high variability mainly with high temperatures and ME-2 not being able to completely determine SVOOA concentrations. These results support the indication that is not an accurate practice to use SVOOA as a target profile when analysing solutions. Trilinear regressions deliver quantitative information about the ratios between combustion tracers and POA. These ratios may be used as a proxy for other urban background sites to estimate POA concentrations.

From analysing heavy and light duty diesel emissions, the main contributor on weekdays was found to be from diesel emissions, particularly LDD emissions. Thus, in order to reduce traffic emissions on weekdays, LDD vehicles should be targeted. For the $PM_{2.5}$ analysis (March-December 2013), the main $PM_1$ contributors to these concentrations were secondary aerosols and BC, which means that $PM_1$ contributors to $PM_{2.5}$ concentrations are related to emissions from combustion activities and also from secondary pollutants produced in the atmosphere.

This study delivers mass spectra and time series of OA sources for a long-term period as well as seasons of the year that maybe used in future studies as TP. Furthermore, the scientific findings provide significant information to strengthen legislation as well as to support health studies in order to improve air quality in the UK.

**Author contributions:** D.C. Green, E. Reyes-Villegas and J.D. Allan designed the project; D.C. Green and M. Priestman operated, calibrated and performed QA of ACSM data; E. Reyes-Villegas performed the data analysis; E. Reyes-Villegas, F. Canonaco, D.C. Green, H. Coe, A.S.H. Prévôt, and J.D. Allan wrote the paper.

**Acknowledgements:** E. Reyes-Villegas is supported with a studentship by the National Council of Science and Technology–Mexico (CONACYT).



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

**Table 1: Sets of target profiles used in the study.**

| a | c | s | w |
|------|------|-------|------|
| BBOA | SFOA | HOA | SFOA |
| HOA | HOA | COA | HOA |
| COA | COA | SVOOA | COA |





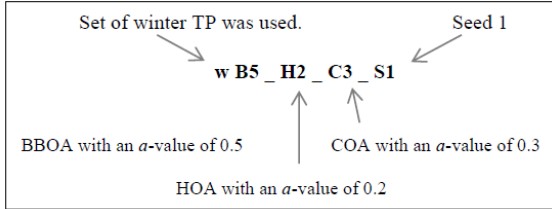

Figure 1: Coding used to identify the different runs.

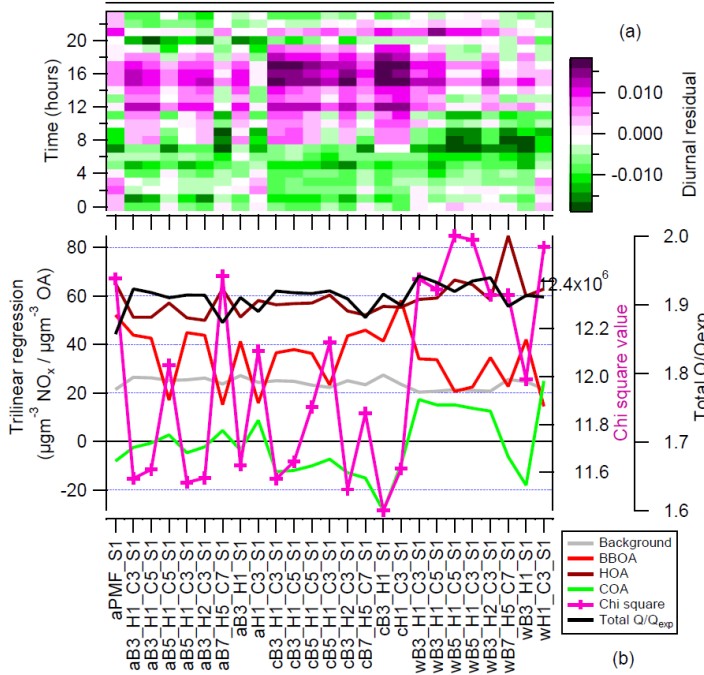


Figure 2 Diurnal residual, y axis represents the 24 hours and x axis the different solutions with a variety of target profiles and *a*-values (a). NO$_x$ trilinear regression for solutions with different target profiles (b). BBOA represents the slope of µgm$^{-3}$ of NO$_x$ per µgm$^{-3}$ of BBOA. The same applies for HOA and COA.

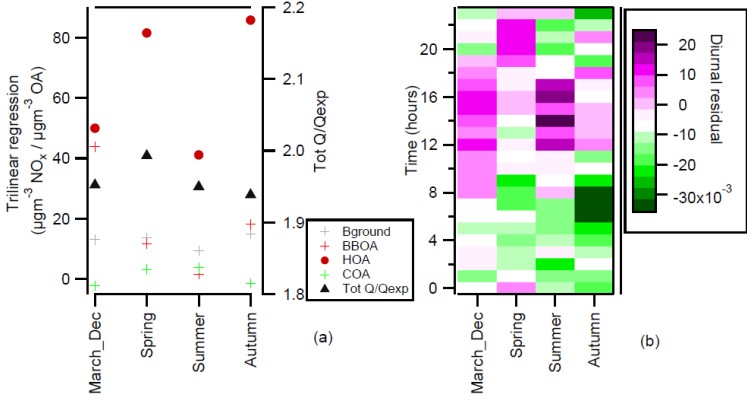


Figure 3: NO$_x$ trilinear regression (a) and diurnal residual (b) for the different analyses.



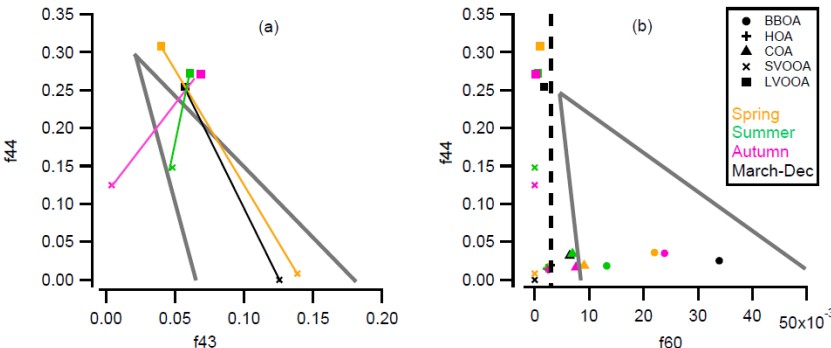

**Figure 4: f44 vs f43 (a) and f44 vs f60 (b) plots for different periods of time.**

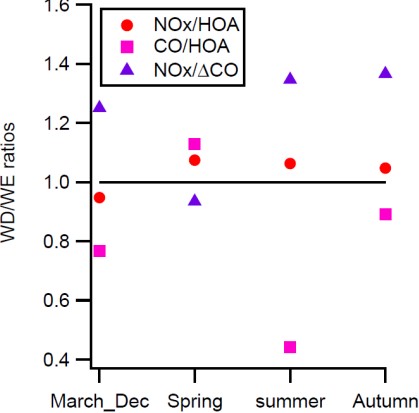

**Figure 5: WD/WE ratios to analyse petrol and diesel contributions.**

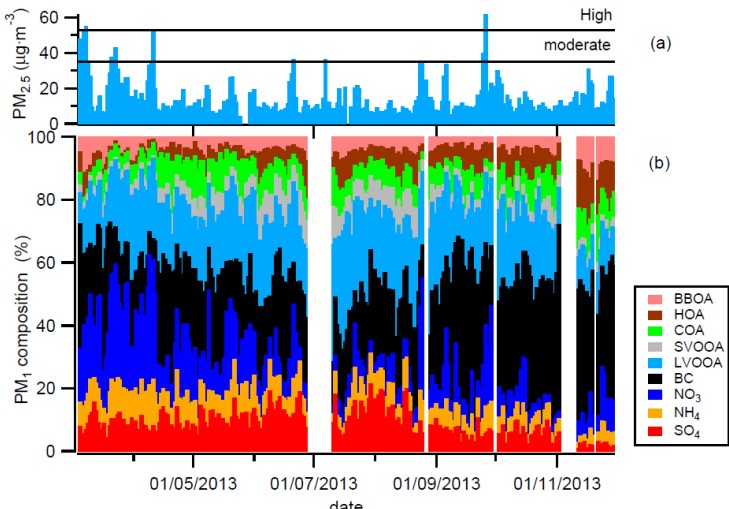

**Figure 6: Daily PM$_{2.5}$ concentrations (a) and daily PM$_1$ composition (b).**