# Peer review of "Organic Aerosol source apportionment in London 2013 with ME-2: exploring the solution space with annual and seasonal analysis"

_Atmospheric Chemistry and Physics, 2016_

## Referee Comment (RC1) · Anonymous Referee #1 · 17 Jun 2016

Review of Reyes-Villegas et al., "Organic aerosol source apportionment in London 2013 with ME-2: exploring the solution space with annual and seasonal analysis

General comments

This paper presents a thorough investigation of the possible factors contributing to OA in London using both traditional PMF and multilinear engine (supervised algorithm). This analysis is useful in that many groups are using PMF and beginning to use ME-2, but so far little validation has been done on the merits or pitfalls of ME-2. Futher, despite the subtlety of the findings and methods, the authors do a reasonable job of guiding the reader through the manuscript. One significant shortcoming of this work is the use of AMS-dervied target profiles to deconvolute ACSM data. While spectra are

similar, important differences have been reported. This is described in detail below. The major concern regarding the use AMS target profiles is that future work will rely on these findings to justify the use of seasonal target profiles (for example). Currently, a number of sections are difficult to read and a few key details are missing. These are relatively minor changes and I recommend this paper for publication following these changes as well as a discussion of the caveats involved in using AMS target profiles with ACSM data.

Specific comments

Abstract line 19: the phrase "seasonal data sets showed a higher seasonal variability" is confusing. After reading the rest of the paper, I think I understand what the authors are saying, but this should be rephrased for clarity. I believe the point is that some sources are seasonal, and therefore the seasons should be evaluated independently rather than running PMF or ME-2 on the full data set.

Line 156: Use of target profiles from AMS and not ACSM may hinder this work, given that Ng et al. 2011 reported a markedly different collection efficiency for HOA and OOA by the ACSM, leading to underestimation of HOA by the ACSM. Further, Ng et al. 2011 reported that when HOA containing particles are externally mixed from OOA particles, this differential collection efficiency could be significant. Do the authors have any insight as to whether this could be important for the London work? At the very least, this shortcoming should be presented clearly at the outset of the work. If ACSM target profiles are available, those would be preferable to test the performance of ME-2.

Line 216: The authors write, "When analysing different solutions from the same dataset (Fig. 2.b), it is possible to observe that the use of different a-values does not imply a high variation, ranging between 1.88-2.2, suggesting that all the solutions are mathematically acceptable." I do not understand the point of this sentence, and I think the main issue is the phrase "does not imply a variation." Please clarify.

Line 220: The authors write "however, [the] PMF solution has a high Chi square and

negative slope for COA (Fig. 2.a), suggesting that this solution is not environmentally acceptable, thus it is necessary to analyse all the different parameters in fig. 2 in order to select the solution that best identifies the OA sources." It is useful to use the trilinear regression to determine if the PMF factors are environmentally realistic. However, the authors should provide more detail here if they intend to use this justification to eliminate the PMF solution. Specifically,for which "y" is the slope negative? BC, CO, NOx or all three? Please indicate here. And, was this true for all the PMF solution rotations under 5 factors? Or just the one with the lowest Q/Qexp? What about seeds? Was this negative slope true for those as well?

Line 292 (first entire paragraph in 4.1.2): I could not parse this paragraph. I think the term "variation" is misused and causes confusion. I understand the value of using the trilinear regression, but this description is nearly incomprehensible. Please revise.

Line 322: This finding, that SVOOA is not always in the same position in the triangle relative to LVOOA seems BIG. Perhaps it could be emphasized more, or earlier? In the abstract?

Table 1: This would be far more useful if the acronyms were avoided or defined.

Figure 2 (a): The color scale is useful to compare residuals, though I question the use of different colors to indicate positive and negative residuals. It would be easier to read the graph if either the best solutions were darkest (since the dark pink looks close to the dark green) OR if the same color was use for both positive and negative. It would also be useful to separate or demarcate the break between "a" and "c" as well as "c" and "w." This would make it clearer that we are comparing those larger groups instead of individual factors.

Technical corrections

Abstract: Please spell out PSI.

Abstract and throughout: Is "interphase" a British form of "interface"? (Line 72)

Line 30: "comprising" should be "composing"

Line 216: insert a space before the sentence beginning "When"

Line 220: should read "the PMF solution" instead of "PMF solution"

Line 298: "All this analysis carried out in the section 4" needs editing for grammar. Perhaps "The analysis presented in Section 4"

Line 380: "found secondary aerosols to dominate high concentrations over the year" should be "found secondary aerosols to be the predominant source of PM over the year"

Line 392: "infer" is misused here, instead the authors could use "support"

---

## Referee Comment (RC2) · Anonymous Referee #2 · 1 Jul 2016

The authors present a source apportionment analysis for 10 months of ACSM measurements in North Kensington, London. ME-2 is used to explore solutions and sensitivity of results to algorithm parameters (seed, number of factors), segregation of data set (whole dataset or by season), and target profiles known a priori. The authors use a regression approach to help determine the final solutions and explore the variability in factor compositions obtained. The authors find that HOA and COA are robust while the other OA to be more variable from the perspective of mass fragment ratios, and that seasonal decomposition results in higher variability (also considering mass fragment ratios). Source apportionment is a difficult problem with a large number of possible solutions, and the authors make a worthwhile contribution in understanding
how to explore this space and the variability in solutions generated from ME-2 using ACSM measurements. The use of trilinear regression is a useful addition for this data set. The results are clearly presented and the topic is suitable for Atmospheric Chemistry and Physics. The manuscript is recommended for publication after the following comments are addressed.

1. Much of the variability in solutions are expressed in terms of fragment ratios. What is the range in OA mass estimated for the set of remaining plausible solutions?

2. The conclusions regarding the superiority of the outlined approach are perhaps stated too strongly by the authors. For instance, in the abstract and conclusions: trilinear regression is said to "objectively determine" the solution, but it is perhaps not truly not objective in that the authors are imposing their prior assumption regarding the nature of cooking emissions and their relation to combustion tracers. And even then, the protocol does not uniquely determine the "best" solution. On this point, it would be useful to plot confidence intervals on the regression coefficients in Figure 2 (and Figure S6) as there are a number of solutions which fulfill this criterion.

Similarly, the statement about ME-2 being "robust" or "using appropriate mass spectra" being important may be revisited. As pointed out by Reviewer 1, there are some concerns about using AMS profiles to constrain ACSM. More generally, introducing source profiles can help reduce the range of solutions compared to PMF. However, it is not made clear in this manuscript that the solutions obtained by this approach are necessarily better than a subset of solutions that can be obtained by PMF (which are derived solely from the ACSM data). Some claims might be made regarding the appropriateness of AMS profiles to the extent that the physical expectations set forth by the authors are met using them (i.e., there are solutions for which COA is not correlated with combustion tracers), but there are still many questions remaining to make too strong a conclusion.

Minor comments:

In Figure 4, it would be helpful to plot points or contours for the range observed in ambient samples for comparison against ME-2 profiles.

Section 4.4: Some discussion of the NO3 and LV-OOA percentages in the text would be useful.

The authors write the essential equations for PMF but not equations for how factor constraints are introduced using the "a-factor" by ME-2, while the rest of the manuscript is dedicated to presenting ME-2 solutions.

Additional minor comments were pointed out by Reviewer 1.
* * *

---

## Referee Comment (RC3) · Anonymous Referee #3 · 6 Jul 2016

This manuscript presents the ME-2 factorization tool to explore the organic aerosol sources in London. The authors use the trilinear regression to compare different ME-2 solutions, and five OA sources were identified including BBOA, HOA, COA, SVOOA, and LVOOA. The authors present the ME-2 solutions and using a-values approach and tested constrained factors. There is high variability in different seasons; however, the authors did not point out the major contributions and specify the variability. The methods and results are presented very well to explore the ME-2 method for the comparisons. The triangle plot of f43: f44 and f44:f60 was applied to further examine the seasonal difference. In general, this manuscript presents lots of scientific results and data analysis, and it's publishable on Atmospheric Chemistry and Physics. Some

concerns and comments are listed.

1. The authors use ME-2 tool to analyze ACSM data to further explore the OA sources and find the best solutions, and the criteria for selecting best solution are using a-values approach, minimizing Q/Qexp, and trilinear regression analysis. However, the criteria of determining best solution are not clearly explained in each analysis (Figure S4~Figure S7).

2. The triangle plot of f43: f44 and f44:f60 was introduced to compare the seasonal differences. The authors also pointed out that it's an oversimplification tool to address the chemical complexity of LVOOA and SVOOA component, and further work is needed to completely address the OA chemistry difference in seasonal changes. Please clarify it or add more details how this triangle plot addresses the results and conclusion. In the abstract, the authors write "the seasonal variability was explored with triangle plots of f43:f44 and f44:f60, with HOA and COA being the most suitable sources to constrain." The COA is mainly characterized by m/z 55 and m/z 57, but here the f60 is low for COA. Is it more appropriate to look at f55 and f57 for COA in seasonal differences?

Minor comments

Line 188 : Equation (4) B and C parameters were not defined.

Line 288-290: The authors write "The summer is overestimated and a strong variation of the source profiles, a situation that ME-2 is not able to capture." If there is a strong variation of the source profile, how does the ME-2 confirm the data accuracy?

Line 296: The authors write " the March-Dec dataset solution does not completely capture". What range of variability for thrilinear regression? The authors write there are seasonal variations that the dataset solution does not completely capture. How does this method completely capture it and avoid failures?

Line 320-322: " The fact that the lines are going in different directions with the seasons of year means that the factorization is identifying different aspects of the chemical

complexity." It seems that the "chemical complexity" is not clear explained why the LVOOA and SVOOA lines are different directions.

Line 330: "The fresh OA during autumn and highly oxygenated BBOA during summer" seems not convinced. It would be good if the authors could provide O/C ratios or any indicators (f44 or f43) that supports the "highly oxygenated BBOA" during summer.

Line 343-345 : Please add reference for this paragraph. The authors use the ratios of NOx/HOA, CO/HOA, and NOx/ △CO to determine the weekdays and weekend ratios. However, there is no strong evidence show that these ratios are best indicators to analyze the impact of diesel or petrol emissions contribution. Also, when the authors conclude the heavy-duty diesel vehicle emissions are possible contributions during weekdays, but it seems not strongly supportive to conclude this.

Line 375 : " The main PM1 contributors to moderate and high PM2.5 concentrations are NO3 and LVOOA." It would be supportive if the authors could provide more information about NO3.

Line 390: In the conclusion, the author write " higher variation mainly in the SVOOA mass spectra and the BBOA; less variability was observed in LVOOA, COA and HOA." The "variability" term was used in this manuscript many times, but there is no clear range for these variations.

Table 1 : As mentioned by reviewer 1, please define clearly for the sets of target profiles.

Technical correction

Figure S1: Please label significant m/z peaks, O/C and H/C ratios in each solution. "(c)" was missing in the end of the caption of Figure S1.

[Figure]

---

## Referee Comment (RC4) · Anonymous Referee #4 · 8 Jul 2016

The work apportions the organic aerosol, measured with an ACSM in North Kensington, London, among five components. Two of the components (BBOA and COA) are clearly sources, while three are definitions of the nature of the organic composition, i.e. not specifically sources. The apportionment is used to identify the relative importance of diesel versus gasoline vehicles to the organic aerosol. The importance of the paper appears to be the detailed evaluation of the approaches including the use of the trilinear regression.

The first art of the paper is well written, giving the reader a clear and detailed discussion of the approaches. Given there are three other reviews, I have only a few minor comments.

1) Line 20 – "detected in the combined March-December dataset."

2) Lines 47-49 – add Takahama et al., Organic Functional Groups in Aerosol Particles from Burning and Non-burning Forest Emissions at a High-Elevation Mountain Site, Atmos. Chem. Phys., 11, 6367–6386, 2011.

3) Line 110 – what is the difference between PM and aerosols?

4) Line 292 – "higher"

5) Line 298 – "The" rather than "All this"

6) Line 317 – I don't see BBOA, HOA or COA in Figure 4a.

7) Lines 318-319 - How do the connecting lines display variability? Are they intended to make the differences between the cluster of LVOOA and the wider spread of SVOOA points more obvious?

8) Lines 328-329 – could cloud processing also contribute to the higher 44 and lower 60?

9) Line 334 – the instead of that.

10) Line 359 – do you mean 1.36 rather than 1.136?

11) Lines 359 – 363 – you use "possible to observe" in a couple of places in this paragraph. I suggest re-writing those segments.

12) Line 369 – "analyzing the aerosol"

13) Line 375 – How do you derive PM1? It is a large unstated assumption if you mean that the ACSM measures PM1. It will measure a larger fraction of the PM1, but it is not a PM1 measurement.

14) Line 379 – Another unsubstantiated claim. Explain how you come to the conclusion that secondary aerosols are the main contributors to PM2.5 from particles smaller than the upper limit of the ACSM. Which components of Figure 6 do you consider secondary

and which do you consider primary?

15 ) Line 394 – higher

---

## Author Comment (AC1) · 8 Sep 2016

Response to comments of referee # 1

Comment 1 from Referee.

Abstract line 19: the phrase "seasonal data sets showed a higher seasonal variability" is confusing. After reading the rest of the paper, I think I understand what the authors are saying, but this should be rephrased for clarity. I believe the point is that some sources are seasonal, and therefore the seasons should be evaluated independently rather than running PMF or ME-2 on the full data set.

author's response

The paragraph was rephrased.

author's changes in manuscript

ME-2 analysis of the seasonal datasets (spring, summer and autumn) showed a higher variability in the OA sources that was not detected in the combined March-December dataset; this variability was explored with the triangle plots f44:f43 f44:f60, where a high variation of SVOOA relative to LVOOA was observed in the f44:f43 analysis. Hence, it was possible to conclude that, when performing source apportionment to long-term measurements important information may be lost and this analysis should be done to short periods of time such as seasonally.

Comment 2 from Referee.

Line 156: Use of target profiles from AMS and not ACSM may hinder this work, given that Ng et al. 2011 reported a markedly different collection efficiency for HOA and OOA by the ACSM, leading to underestimation of HOA by the ACSM. Further, Ng et al. 2011 reported that when HOA containing particles are externally mixed from OOA particles, this differential collection efficiency could be significant. Do the authors have any insight as to whether this could be important for the London work? At the very least, this shortcoming should be presented clearly at the outset of the work. If ACSM target profiles are available, those would be preferable to test the performance of ME-2.

author's response

This situation was taken into consideration when first defining the way to analyse this dataset. For this reason, it was decided to run ME-2 with different a-values and target profiles from a variety of studies, in order to explore the solution space and select the solution that best identified the OA sources, according to the statistical tests applied. As shown in section 4, the use of AMS mas spec as target profiles, while giving a wide range of possible solutions with different target profiles, delivered successful results. The ACSM was specifically designed to deliver mass spectra that were equivalent to

the AMS and so there is every reason to expect that using AMS spectra is appropriate, as they are not just comparable but should have a higher signal-to-noise ratio. Given that the purpose of this work is to systematically assess the suitability of target profiles, it would be remiss of us not to consider AMS-generated target profiles, especially considering there are more of these available, including ones obtained from the same site. Frolich, et al. (2015) used AMS mass spectra as target profiles to deconvolve OA measured with 14 ACSM and one AMS during an intercomparison, being able to successfully separate 4 different sources by partially constraining COA and HOA.

A new section in the supplement (section S3) has been added with a PMF analysis with different f-peaks in order to show how the PMF solution was chosen for the further comparison with ME2 solutions showing that ME-2 analysis improved the OA source identification compared to PMF solutions. For all these reasons we do not consider that using AMS TP has a negative effect on the ME-2 source apportionment.

author's changes in manuscript

This paragraph will be added at the end of the first paragraph in section 2.2.1

The ACSM was specifically designed to deliver mass spectra that were equivalent to the AMS; with the AMS having a higher signal to noise ratio, it is expected the use of its mass spectra as TP to be appropriate. Moreover, we consider AMS-generated TP to be convenient to use especially considering there are more of these available, including the ones obtained from the same site. In this study, the suitability of different TP will be systematically assessed in the determination of OA sources using a wide range of a-values.

Comment 3 from Referee.

Line 216: The authors write, "When analysing different solutions from the same dataset (Fig. 2.b), it is possible to observe that the use of different a-values does not imply a high variation, ranging between 1.88-2.2, suggesting that all the solutions are mathematically acceptable." I do not understand the point of this sentence, and I think the main issue is the phrase "does not imply a variation." Please clarify.

author's response

The idea here is that Q/Qexp in isolation does not identify the best solution and more parameters need to be analysed in order to determine the solution that separates better the different OA sources. The variation mentioned is comparing these Q/Qexp values with the ones in the literature (section 4.1.1)

author's changes in manuscript

For an ideal solution a Q/Qexp value of 1.0 would be expected. However, there is not a standard criterion to define a satisfactory Q/Qexp value as a certain amount of 'model error' will cause it to be systematically higher than unity (Ulbrich et al., 2009).When comparing different solutions from the same dataset (Fig. 2.b), it is possible to observe that there is not a significant variation on the Q/Qexp (ranging between 1.88-2.2) when using different a-values, suggesting that all the solutions are mathematically acceptable. The unconstrained solution is the one with the lowest total Q/Qexp with a value of 1.88, which is expected, as PMF calculates the solution by minimising this value; however, the PMF solution has a high Chi square and negative slope for COA (Fig. 2.a), implying that this solution is not environmentally acceptable, thus it is necessary to analyse all the different parameters in fig. 2 in order to select the solution that best identifies the OA sources.

Comment 4 from Referee.

Line 220: The authors write "however, [the] PMF solution has a high Chi square and negative slope for COA (Fig. 2.a), suggesting that this solution is not environmentally acceptable, thus it is necessary to analyse all the different parameters in fig. 2 in order to select the solution that best identifies the OA sources." It is useful to use the trilinear regression to determine if the PMF factors are environmentally realistic. However,

the authors should provide more detail here if they intend to use this justification to eliminate the PMF solution. Specifically,for which "y" is the slope negative? BC, CO, NOx or all three? Please indicate here. And, was this true for all the PMF solution rotations under 5 factors? Or just the one with the lowest Q/Qexp? What about seeds? Was this negative slope true for those as well?

author's response

The explanation about how to determine the best solution has been explained with further details. In general: uconstrained runs with three seeds were performed with different number of factor in order to determine the most suitable number of factors, after defining 5 factor to be the most adequate number of factors, unconstrained analysis exploring the f-peak from -1 to 1 with steps of 0.1 (Fig. S4, S6, S8 and S10) was performed to determine the PMF solution to be compared with the ME-2 solutions.

author's changes in manuscript

Section 3.1.1 Unconstrained runs with f-peak = 0 and 3 different seeds were performed in order to determine the number of OA sources, being five (BBOA, HOA, COA, SVOOA, LVOOA) the most adequate number of sources (Fig. S1.b) as it was possible to split the SOA into SVOOA and LVOOA. Further unconstrained analysis was performed by running five factor solutions with different f-peaks, from -1 to 1 with steps of 0.1 (Fig. S4) in order to select the PMF solution to be compared with the ME-2 analysis.

Section 3.1.3 Diurnal concentrations for all the solutions (Supplement S3) were analysed to determine the main sources. Here, it was possible to observe that solutions with undesirable outputs in the residual, total Q/Qexp and/or trilinear regression were likely to have mixed diurnal concentrations between two sources. For example in the case of "c" TP solutions, CO and BC trilinear regressions (Fig. S5.a and S5.b) show better COA slopes with values close to zero, however due to the high diurnal residual (Fig. 2.a) and HOA with high concentrations during the evening (Fig. S5.c), suggesting

mixing with BBOA, "c" TP solutions are not considered acceptable solutions.

Comment 5 from Referee.

Line 292 (first entire paragraph in 4.1.2): I could not parse this paragraph. I think the term "variation" is misused and causes confusion. I understand the value of using the trilinear regression, but this description is nearly incomprehensible. Please revise.

author's response

The paragraph has been edited in the manuscript

author's changes in manuscript

Looking at the trilinear outputs for the different periods of time analysed (Figure 3.a), NOx/HOA ratios present higher variability with values of 50.0 for March-Dec, 81.0 for spring, 41.0 for summer and 85.5 for autumn. The different NOx/BBOA and NOx/HOA slopes for spring, summer and autumn suggest that when looking at March-Dec solution only, there are seasonal variations that this solution does not completely capture. With regard to background and NOx/COA slopes, they are well identified and relatively constant over the different periods of time analysed.

Comment 6 from Referee.

Line 322: This finding, that SVOOA is not always in the same position in the triangle relative to LVOOA seems BIG. Perhaps it could be emphasized more, or earlier? In the abstract?

author's response

This finding has been added to the abstract.

author's changes in manuscript

ME-2 analysis of the seasonal datasets (spring, summer and autumn) showed a higher variability in the OA sources that was not detected in the combined March-December

dataset; this variability was explored with the triangle plots f44:f43 f44:f60, where a high variation of SVOOA relative to LVOOA was observed in the f44:f43 analysis. Hence, it was possible to conclude that, when performing source apportionment to long-term measurements, this analysis should be done to short periods of time such as seasonally.

Comment 7 from Referee.

Table 1: This would be far more useful if the acronyms were avoided or defined.

author's response

The acronyms description is shown in the section 2.2.1 where the table will be displayed in the final manuscript.

author's changes in manuscript

None.

Comment 8 from Referee.

Figure 2 (a): The color scale is useful to compare residuals, though I question the use of different colors to indicate positive and negative residuals. It would be easier to read the graph if either the best solutions were darkest (since the dark pink looks close to the dark green) OR if the same color was use for both positive and negative. It would also be useful to separate or demarcate the break between "a" and "c" as well as "c" and "w." This would make it clearer that we are comparing those larger groups instead of individual factors.

author's response

The use of different colours for negative and positive residuals is useful in the way that it tells the reader if the solution is overestimated (negative residual) or underestimated (positive residual). About using a break to separate solutions with different target profiles is a reasonable suggestion; this will be modified in the updated manuscript.

author's changes in manuscript

Figure 2 Diurnal residual, y axis represents the 24 hours and x axis the different solutions with a variety of target profiles and a-values (a). NOx trilinear regression for solutions with different target profiles (b). BBOA represents the slope of $\mu$gm-3 of NOx per $\mu$gm-3 of BBOA. The same applies for HOA and COA. Whiskers represent the 95% confidence interval.

All the technical corrections were taken into account and modifications were done in the manuscript

[Figure]

**Fig. 1.** Figure 2

---

## Author Comment (AC2) · 9 Sep 2016

Response to comments of referee # 2

Comment 1 from Referee.

Much of the variability in solutions are expressed in terms of fragment ratios. What is the range in OA mass estimated for the set of remaining plausible solutions?

author's response

The OA mass estimated for all the different solutions was practically the same, with summer being the period with less OA mass estimated (90%) and the other periods

with more than 95% of mass estimated from the total OA concentrations. Figure S12 has been added to the supplement.

author's changes in manuscript

at the end of section 3.2 It is worth to mention that all plausible solutions deconvolved a high percentage of the total OA mass (Fig. S12), with summer being the period with less OA mass estimated (90%) and the other periods with more than 95% of mass estimated from the total OA concentrations.

Figure S1 added to the supplement.

Comment 2 from Referee.

The conclusions regarding the superiority of the outlined approach are perhaps stated too strongly by the authors. For instance, in the abstract and conclusions: trilinear regression is said to "objectively determine" the solution, but it is perhaps not truly not objective in that the authors are imposing their prior assumption regarding the nature of cooking emissions and their relation to combustion tracers. And even then, the protocol does not uniquely determine the "best" solution. On this point, it would be useful to plot confidence intervals on the regression coefficients in Figure 2 (and Figure S6) as there are a number of solutions which fulfill this criterion.

Similarly, the statement about ME-2 being "robust" or "using appropriate mass spectra" being important may be revisited. As pointed out by Reviewer 1, there are some concerns about using AMS profiles to constrain ACSM. More generally, introducing source profiles can help reduce the range of solutions compared to PMF. However, it is not made clear in this manuscript that the solutions obtained by this approach are necessarily better than a subset of solutions that can be obtained by PMF (which are derived solely from the ACSM data). Some claims might be made regarding the appropriateness of AMS profiles to the extent that the physical expectations set forth by the authors are met using them (i.e., there are solutions for which COA is not corre-

lated with combustion tracers), but there are still many questions remaining to make too strong a conclusion.

author's response

By 'objective', we wished to convey that the tests being applied to the data were being performed in a quantitative and consistent manner with as little subjective inspection of the results as possible. This is to distinguish from many interpretations of multivariate analyses, where there is often a tendency to approve solution sets based on criteria that are defined a posteriori. However, we concede that describing the tests as 'objective' may not have been appropriate. While the tests rely on the model that the factors behave according to a certain way relative to the tracers, we deem this appropriate if the factors are to be specifically assigned to physical sources such as traffic, cooking, etc. However, this is one of a number of tests, including Q/Qexp, diurnal residuals, diurnal concentrations and trilinear regression (Chi-square and slopes) for different number of seeds and a-values. By 'best' solution, we mean that this is the solution that we can deem optimal according to the results of the tests we have available. Confidence intervals were added to the trilinear regression plots.

About using AMS mas spec: This situation was taken into consideration when first defining the way to analyse this dataset. For this reason, it was decided to run ME-2 with different a-values and target profiles from a variety of studies in order to explore the solution space and select the solution that best identified the OA sources, according to the statistical tests applied. Frolich, et al. (2015) used AMS mass spectra as target profiles to deconvolve OA measured with 13 ACSM during an intercomparison, being able to successfully separate 4 different sources by partially constraining COA and HOA. While a high variability was found on the LVOOA mass spectra, minor effect was observed on the actual time series. PMF solutions were explored for different f-peaks from -1 to 1 with steps of 0.1 (Fig. S4, S6, S8 and S10) to show how one PMF solution was chosen to be compared with the ME-2 solutions and the improvement of the ME-2 solution compared with the unconstrained run. Fig. S4 is shown in the following

paragraphs.

author's changes in manuscript

The sentence in the abstract has been modified as follows: "A strategy to explore the solution space is proposed, where the solution that best describes the organic aerosol (OA) sources is determined according to the systematic application of predefined statistical tests. This includes trilinear regression, which proves to be a useful tool to compare different ME-2 solutions."

Sentences were paraphrased; when the word "objectively" was mentioned, it was changed in the manuscript for "systematically", for example:

. . .new strategies to objectively explore the solutions are needed.

was paraphrased for:

. . .new strategies to systematically explore the solutions are needed.

Sentences were also paraphrased in the manuscript. When we are indicating "the best solution" that deconvolves the OA sources, for example:

. . .consistent with the diurnal residual analysis that the best solution is with the solutions constrained with "a" target profiles.

. . .consistent with the diurnal residual analysis that the best solution, according to the statistical tests applied, is with the solutions constrained with "a" target profiles.

This paragraph will be added at the beginning of the first paragraph in section 2.2.1

The ACSM was specifically designed to deliver mass spectra that were equivalent to the AMS; with the AMS having a higher signal to noise ratio, it is expected the use of its mass spectra as TP to be appropriate. Moreover, we consider AMS-generated TP to be convenient to use especially considering there are more of these available, including the ones obtained from the same site. In this study, the suitability of different

TP will be systematically assessed in the determination of OA sources using a wide range of a-values.

Section 2.3 fourth paragraph. The following considerations should be taken into account: The slopes and intercepts should be positive as they represent air pollutant concentrations and the slope D is used as a validation parameter which should be close to zero, due to its low contribution to BC, NOx and CO, owing to the fact that most cooking in the UK uses electricity or natural gas as a source of heat (DECC, 2015;NAEI, 2016).

Section 3.1.1 Unconstrained runs with f-peak = 0 and 3 different seeds were performed in order to determine the number of OA sources, being five (BBOA, HOA, COA, SVOOA, LVOOA) the most adequate number of sources (Fig. S1.b) as it was possible to split the SOA into SVOOA and LVOOA. Further unconstrained analysis was performed by running five factor solutions with different f-peaks, from -1 to 1 with steps of 0.1 (Fig. S4) in order to select the PMF solution to be compared with the ME-2 analysis.

Paragraph added to the supplement: PMF runs were performed, for the March-December period, from fpeak -1 to 1 with steps of 0.1. Figure 4 shows the comparison of the runs that converged (some of the fpeaks did not converge) in order to determine the PMF solution that better identified the OA sources to be compared to the ME-2 solutions. Run number 4 is chosen to be the best solution, according to the statistical tests applied, with low diurnal residual and positive COA for CO and BC trilinear regressions.

Minor comments Comment 1 from Referee.

In Figure 4, it would be helpful to plot points or contours for the range observed in ambient samples for comparison against ME-2 profiles.

author's response f43, f44 and f60 of the OA measurements were added to figure 4.

author's changes in manuscript

figure 4.

Comment 2 from Referee. Section 4.4: Some discussion of the NO3 and LV-OOA percentages in the text would be useful.

author's response

The paragraph has been modified.

author's changes in manuscript

Considering that PM1 is composed mainly of OA, SO4, NO3, NH4 and BC, it is possible to analyse the PM1 composition during PM2.5 high concentrations (Fig. 6.b). Episodes with moderate and high PM2.5 concentrations were observed with low wind speeds (Fig. S8), being NO3 and LVOOA the main PM1 contributors. High NO3 concentrations were observed during spring as found in a previous study performed by Young et al. (2015a) who determined that NO3 concentrations in spring depend on air mass trajectory, precursors and meteorology. Different contributions from OA sources were identified: in the episode in March, high BBOA concentrations were observed, whereas during the episodes in April and September, higher concentrations of LVOOA were detected.

Comment 3 from Referee.

The authors write the essential equations for PMF but not equations for how factor constraints are introduced using the "a-factor" by ME-2, while the rest of the manuscript is dedicated to presenting ME-2 solutions.

author's response

The basic equation to apply the a-value was added to the manuscript. author's changes in manuscript

The equation 4 was applied using different target profiles (gi) and a range of a-values (a) to constrain OA sources in different runs (gi,run).

g_(i,run)=g_i±a*g_i (4)

(Equation 4 with proper symbols is shown at the end of the attached file with figures).

[Figure]

**Figure S1: OA concentrations and proportions of the different OA sources to the total OA. March-Dec (a), spring (b), Summer (c) and autumn (d).**

**Fig. 1.** Figure S1

[Figure]

Figure S1 : NOx, CO and BC trilinear regression (a, b, c), diurnal residual (d), diurnal concentrations (e) and solution list for March-Dec PMF analysis (f).

**Fig. 2.** Figure S2

[Figure]

Figure 4: f44 vs f43 (a) and f44 vs f60 (b) plots for different periods of time.

The equation 4 was applied using different target profiles ($g_i$) and a range of a-values ($a$) to constrain OA sources in different runs ($g_{i,run}$).

$$g_{i,run} = g_i \pm a * g_i \qquad (4)$$

**Fig. 3.** Figure S4_equation4

---

## Author Comment (AC3) · 9 Sep 2016

Response to comments of referee # 3

Comment 1 from Referee.

The authors use ME-2 tool to analyze ACSM data to further explore the OA sources and find the best solutions, and the criteria for selecting best solution are using avalues approach, minimizing Q/Qexp, and trilinear regression analysis. However, the criteria of determining best solution are not clearly explained in each analysis (Figure S4_Figure S7).

author's response

The description has been further explained in supplement S3. Moreover, a description how the PMF was chosen has been added with the Figure S4.

author's changes in manuscript

S3. Analysis to determine the best solution for the different periods of time.

PMF runs were performed, for the March-December period, from fpeak -1 to 1 with steps of 0.1. Figure 4 shows the comparison of the runs that converged (some of the fpeaks did not converge) in order to determine the PMF solution that better identified the OA sources to be compared to the ME-2 solutions. Run number 4 is chosen to be the best solution, according to the statistical tests applied, with low diurnal residual and positive COA for CO and BC trilinear regressions.

Figure S5 shows the analysis performed to determine the best solution for the March-December period. As mentioned in the main text of this paper, "c" and "w" target profiles (TP) show the less desirable results; "c" TP show a high positive residual (Figure 2.a) and "w" TP show a high chi-square and COA slope. (Figures C1.a and S4.b). From the "a" TP, aB3_H2_C3_S1 solution is chosen to present the best results from this analysis due to COA slope close to zero for NOx (Figure 2.b) and CO (Figure S5.a) trilinear regression and low diurnal residual (Figure 2.a).

Figure S6 shows the PMF analysis for the spring period. All solutions show similar diurnal concentrations with negative COA slope fo the three trilinear regressions. Solutions 2 and 3 have the lower Q/Qexp, Solution 3 was chosen to be compared with ME-2 solutions.

Figure S7 shows the analysis performed to determine the best solution for spring period. Solutions with "a" and "c" TP show the less desirable results with negative slopes for COA and high chi-square in the trilinear regression (Figures S5.a, S5.b and S5.c), "c" TP also show high diurnal residuals. The solution wB3_H1_C3_S1 is chosen to present the best results from this analysis with low chi-square and diurnal residuals.

Figure S8 shows the PMF analysis for the summer period. Solution 4 has a high Q/Qexp but as it shows a COA slope close to zero in the three trilinear analyses and a low diurnal residual compared to the other PMF solutions, it has been chosen to be compared with ME-2 solutions.

Figure S9 shows the analysis performed to determine the best solution for summer period. Solutions with "c" and "s" TP show the less desirable results. "s" TP show low chi-square values, however, they present high negative residuals in the morning and at night. "c" TP show a high positive residual around 15:00-18:00 hrs. The solution aB5_H1_C3_S1 is chosen to present the best results from this analysis dueto the low diurnal residual, COA slope close to zero and the low BBOA slope in the NOx, BC and COA trilinear regressions (Figures S9.a, S9.b and S9.c).

Figure S10 shows the PMF analysis for the autumn period. Solution 4 has been the chosen solution to be compared with ME-2 solutions because of its low Q/Qexp and a COA slope close to zero for the NOx trilinear regression and a lower diurnal residual compared to the other PMF solutions.

Figure S11 shows the analysis performed to determine the best solution for autumn period. Solutions with "a" TP show the less favourable Chi square results in the three trilinear regression figures (Figures S11.a, S11.b and S11.c). wB3_H1_S1 solution is chosen to present the best results from this analysis with low chi-squares and COA slope close to zero in the trilinear regression with NOx (Figures S11.a).

Comment 2 from Referee.

The triangle plot of f43: f44 and f44:f60 was introduced to compare the seasonal differences. The authors also pointed out that it's an oversimplification tool to address the chemical complexity of LVOOA and SVOOA component, and further work is needed to completely address the OA chemistry difference in seasonal changes. Please clarify it or add more details how this triangle plot addresses the results and conclusion. In the abstract, the authors write "the seasonal variability was explored with triangle plots of

f43:f44 and f44:f60, with HOA and COA being the most suitable sources to constrain." The COA is mainly characterized by m/z 55 and m/z 57, but here the f60 is low for COA. Is it more appropriate to look at f55 and f57 for COA in seasonal differences?

author's response The paragraph in the abstract has been modified. The main idea here is to describe the seasonal variation mainly observed in SVOOA rather than mentioning the most suitable sources to constrain. More details about the f44:f43 variation in the triangle plot have been added to the manuscript. author's changes in manuscript abstract: ME-2 analysis of the seasonal datasets (spring, summer and autumn) showed a higher variability in the OA sources that was not detected in the combined March-December dataset; this variability was explored with the triangle plots f44:f43 f44:f60, where a high variation of SVOOA relative to LVOOA was observed in the f44:f43 analysis. Hence, it was possible to conclude that, when performing source apportionment to long-term measurements important information may be lost and this analysis should be done to short periods of time such as seasonally. Section 4.2 This analysis shows that the factors derived for SOA do not always conform to the model of LVOOA and SVOOA proposed by Jimenez et al. (2009). Furthermore, the fact that the lines are going in different directions with the seasons of year means that the factorisation is identifying different aspects of the chemical complexity, as LVOOA and SVOOA rather than being originated from primary emissions are part of continuous physico-chemical processes involving gases, aerosols and meteorological parameters among others. This serves to highlight that a 2-component model (LVOOA and SVOOA) is an oversimplification of a complex chemical system as concluded by Canonaco et al. (2015) who found significant f44 vs f43 difference for summer and winter analyses. Minor comments Comment 1 from Referee. Line 188 : Equation (4) B and C parameters were not defined.

author's response B and C parameters have been defined in the manuscript. author's changes in manuscript B, C and D slopes represent the contribution of BBOA, HOA and COA to "Y" and the intercept A is representative of the "Y" background concentration.

Comment 2 from Referee. Line 288-290: The authors write "The summer is overesti-mated and a strong variation of the source profiles, a situation that ME-2 is not able to capture." If there is a strong variation of the source profile, how does the ME-2 confirm the data accuracy?

author's response The comparison here is with the diurnal residual for the different analyses (March-Dec, Spring, Summer and Autumn). As mentioned in the manuscript, PMF considers the sources to be constant over time which is not true and more in daytime during summer when more photochemistry is happening, affecting the source apportionment. We can see this effect on the diurnal residual with positive concen-trations between 12:00 to 17:00 UTC. However, the concentrations in the residuals of 0.018 are low compared to the diurnal concentrations of the different sources (Fig S6e). author's changes in manuscript It is in the diurnal residual where we can observe a high variation (Fig. 3.b), with autumn proving to be the most overestimated with neg-ative residuals of -0.033 $\mu$g.m-3 mainly in the morning and at night. On the other hand, summer shows to be the most underestimated solution with values of 0.018 $\mu$g.m-3 particularly between midday and 17:00 UTC. The fact that summer is underestimated from 12:00 to 17:00 UTC is probably related to the increase on photochemical activity, a situation that ME-2 is not able to capture as considers the mass spectra to remain constant over the period analysed. It is important to notice that these diurnal residuals of 0.03 $\mu$g.m-3 or less are low compared with diurnal concentrations of the OA sources, which concentrations ranged 0.1-0.6 $\mu$g.m-3.

Comment 3 from Referee. Line 296: The authors write " the March-Dec dataset solu-tion does not completely capture". What range of variability for thrilinear regression? The authors write there are seasonal variations that the dataset solution does not com-pletely capture. How does this method completely capture it and avoid failures?

author's response

The paragraph has been edited to better explain the seasonal variations. It is in all the

section 4.1 where we show how seasonal solutions present different values of Q/Qexp, residuals and POA slopes suggesting that there is a seasonal dependency on the different OA sources which would not be possible to determine when running ME-2 for long periods of time (March-Dec).

author's changes in manuscript Looking at the trilinear outputs for the different periods analysed (Figure 3.a), HOA slopes present higher variability with values of 50.0 for March-Dec, 81.0 for spring, 41.0 for summer and 85.5 for autumn. The different BBOA and HOA slopes for spring, summer and autumn suggest that when looking at March-Dec solution only, there are seasonal variations, perhaps affected by changes on the inhabitants' daily activities (i.e. domestic heating) and meteorological conditions, that the March-Dec solution does not completely capture. With regard to COA slopes and background concentrations, they are well identified and relatively constant over the different periods analysed. Comment 4 from Referee. Line 320-322: " The fact that the lines are going in different directions with the seasons of year means that the factorization is identifying different aspects of the chemical complexity." It seems that the "chemical complexity" is not clear explained why the LVOOA and SVOOA lines are different directions.

author's response

The paragraph was edited adding explantion about what we mean of "chemical complexity" author's changes in manuscript Furthermore, the fact that the lines are going in different directions with the seasons of year means that the factorisation is identifying different aspects of the chemical complexity, as LVOOA and SVOOA rather than being originated from primary emissions are part of continuous physicochemical processes involving gases, aerosols and meteorological parameters among others. This serves to highlight that a 2-component model (LVOOA and SVOOA) is an oversimplification of a complex chemical system as concluded by Canonaco et al. (2015) who found significant f44 vs f43 difference for summer and winter analyses. Comment 5 from Referee. Line 330: "The fresh OA during autumn and highly oxygenated BBOA during summer"

seems not convinced. It would be good if the authors could provide O/C ratios or any indicators (f44 or f43) that supports the "highly oxygenated BBOA" during summer. author's response paragraph has been edited on the manuscript author's changes in manuscript BBOA evolution has been frequently observed with high f44 and low f60 values due to due to aging, oxidation and cloud processing (Huffman et al., 2009;Cubison et al., 2011). Thus, it was possible to obtain a variety of BBOA for the different seasons of the year, ranging from a fresh BBOA with a high f60 during autumn to a more oxidised BBOA with a low f60 during summer.

Comment 6 from Referee. Line 343-345 : Please add reference for this paragraph. The authors use the ratios of NOx/HOA, CO/HOA, and NOx/ _CO to determine the weekdays and weekend ratios. However, there is no strong evidence show that these ratios are best indicators to analyze the impact of diesel or petrol emissions contribution. Also, when the authors conclude the heavy-duty diesel vehicle emissions are possible contributions during weekdays, but it seems not strongly supportive to conclude this.

author's response

The paragraph has been edited and more references have been added.

author's changes in manuscript

Diesel emits higher NOx and HOA concentrations compared to petrol, while petrol emits higher concentrations of CO, according to the National Atmospheric Emissions inventory (NAEI, 2016), during 2014 the emission factors (units in kilotonnes of pollutant per Megatonne of fuel used) were: 11-12 for diesel and 1.9-4.3 for petrol in the case of NOx and 2.4-5.6 for diesel and 11-50 for petrol in the case of CO. Moreover, there are variations between Light Duty Diesel (LDD) and Heavy Duty Diesel (HDD) emissions (LAEI, 2013), with LDD emitting higher NOx concentrations and HDD emitting higher HOA concentrations.

It is possible to qualitatively analyse the impact of different fuels on air pollution by

looking at weekday/weekend ratios (WD/WE), as previously done in several studies (Bahreini et al., 2012;Tao and Harley, 2014;DeWitt et al., 2015) and stating the hypothesis that different fuels will have different pollutant contribution during the week.

Comment 7 from Referee.

Line 375 : " The main PM1 contributors to moderate and high PM2.5 concentrations are NO3 and LVOOA." It would be supportive if the authors could provide more information about NO3.

author's response

The paragraph has been edited.

author's changes in manuscript

Considering that PM1 is composed mainly of OA, SO4, NO3, NH4 and BC, it is possible to analyse the PM1 composition during PM2.5 high concentrations (Fig. 6.b). Episodes with moderate and high PM2.5 concentrations were observed with low wind speeds (Fig. S8), being NO3 and LVOOA the main PM1 contributors. High NO3 concentrations were observed during spring as found in a previous study performed by Young et al. (2015a) who determined that NO3 concentrations in spring depend on air mass trajectory, precursors and meteorology. Different contributions from OA sources were identified: in the episode in March, high BBOA concentrations were observed, whereas during the episodes in April and September, higher concentrations of LVOOA were detected.

Comment 8 from Referee.

Line 390: In the conclusion, the author write " higher variation mainly in the SVOOA mass spectra and the BBOA; less variability was observed in LVOOA, COA and HOA." The "variability" term was used in this manuscript many times, but there is no clear range for these variations.

author's response

The paragraph has been rewritten to better explain these variations.

author's changes in manuscript

ME-2 proved to be a robust tool to deconvolve OA sources. This study highlighted the importance of using appropriate mass spectra as target profiles and a-values when exploring the solution space. With the implementation of new techniques to compare different solutions, it was possible to systematically determine the solution with the best separation of OA sources, mathematically and environmentally speaking. The comparison carried out between the solution for the March-December dataset and the seasonal solutions showed high variations mainly in the SVOOA and the BBOA sources, with wide range of f44:f43 values for SVOOA (Fig. 4.a) and f60 values ranging from 13x10-3 for summer to 24x10-3 for autumn (Fig. 4.b). These variations support the importance of running ME-2 during periods of time where weather conditions and Âñemissions from human activities are less variable, such as seasonal analyses.

Comment 8 from Referee.

Table 1 : As mentioned by reviewer 1, please define clearly for the sets of target profiles.

author's response

The set of target profiles were previously described in the section 2.2.1 of the manuscript were this table will be part of in the final version. We do not consider necessary to repeat the description in the table.

author's changes in manuscript

None.

Technical correction

Figure S1: Please label significant m/z peaks, O/C and H/C ratios in each solution.

"(c)" was missing in the end of the caption of Figure S1.

author's response

Figure S1 has been updated.

————————————————————

[Figure]

[Figure]

Figure S4 : NOx, CO and BC trilinear regression (a, b, c), diurnal residual (d), diurnal concentrations (e) and solution list for March-Dec PMF analysis (f).

| # | run | fpeak |
|---|-----|-------|
| 1 | PMF_5F_10neg | -1 |
| 2 | PMF_5F_5neg | -0.5 |
| 3 | PMF_5F_1neg | -0.1 |
| 4 | PMF_5F_0 | 0.0 |
| 5 | PMF_5F_1pos | 0.1 |
| 6 | PMF_5F_3pos | 0.3 |
| 7 | PMF_5F_5pos | 0.5 |
| 8 | PMF_5F_8pos | 0.8 |

**Fig. 1.** Figure S4

[Figure]

Figure S6: NOx, CO and BC trilinear regression (a, b, c), diurnal residual (d), diurnal concentrations (e) and solution list for spring PMF analysis (f).

**Fig. 2.** Figure S6

[Figure]

**Figure S8:** NOx, CO and BC trilinear regression (a, b, c), diurnal residual (d), diurnal concentrations (e) and solution list for summer PMF analysis (f).

**Fig. 3.** Figure S8

[Figure]

Figure S10: NOx, CO and BC trilinear regression (a, b, c), diurnal residual (d), diurnal concentrations (e) and solution list for autumn PMF analysis (f).

| # | run | fpeak |
|---|-----|-------|
| 1 | PMF_5F_6neg | -0.6 |
| 2 | PMF_5F_5neg | -0.5 |
| 3 | PMF_5F_1neg | -0.1 |
| 4 | PMF_5F_0 | 0.0 |
| 5 | PMF_5F_1pos | 0.1 |
| 6 | PMF_5F_3pos | 0.3 |
| 7 | PMF_5F_5pos | 0.5 |
| 8 | PMF_5F_6pos | 0.6 |

**Fig. 4.** Figure S10

[Figure]

**Figure S12: OA concentrations and proportions of the different OA sources to the total OA. March-Dec (a), spring (b), Summer (c) and autumn (d).**

**Fig. 5.** Figure S12

---

## Author Comment (AC4) · 9 Sep 2016

Response to comments of referee # 4

Comment 1 from Referee.

Line 20 – "detected in the combined March-December dataset."

author's response

Phrase has been edited as suggested.

author's changes in manuscript

detected in the combined March-December dataset.

Comment 2 from Referee.

Lines 47-49 – add Takahama et al., Organic Functional Groups in Aerosol Particles from Burning and Non-burning Forest Emissions at a High-Elevation Mountain Site, Atmos. Chem. Phys., 11, 6367–6386, 2011.

author's response

The citation has been added to the manuscript.

author's changes in manuscript

In particular, the Aerosol Chemical Speciation Monitor (ACSM), which has been recently developed (Ng et al., 2011), has been used to carry out long-term measurements of non-refractory submicron aerosols around the world, for instance an industrial-residential area in Atlanta, Georgia (Budisulistiorini et al., 2013), a high-elevation mountain in Canada (Takahama et al., 2011), at background locations in South Africa, (Vakkari et al., 2014) and Spain (Minguillón et al., 2015a;Ripoll et al., 2015), a semi-rural site in Paris (Petit et al., 2015) and at an urban background site in Switzerland (Canonaco et al., 2015).

Comment 3 from Referee.

Line 110 – what is the difference between PM and aerosols?

author's response

PM refers specifically to solid and liquid particles and aerosols are these liquid and solid particles in a gas (usually air). There are not main differences between these two terms and PM is used when the particle size is taken into consideration. The intention with this paragraph was to show different studies carried out at the North Kensington site and the particle size is important to mention to compare different studies. The paragraph has been rewritten to compare the different studies based on the particle

size.

author's changes in manuscript

Different studies have been carried out at this site such as analysis of elemental and organic carbon concentrations in offline measurements of particulate matter with a diameter less than 10 micrometres (PM10) (Jones and Harrison, 2005), PM10 and NOx association with wind speed (Jones et al., 2010), properties of nanoparticles (Dall'Osto et al., 2011), PM10 and PM2.5 (Liu and Harrison, 2011) and in aerosol chemical composition (Beccaceci et al., 2015) in the atmosphere. The first long-term study of the behaviour of non-refractory inorganic and organic aerosols (PM1) at the North Kensington site was carried out analysing cToF-AMS data collected from January 2012 to January 2013 (Young et al., 2015a) where source apportionment analysis was carried out applying unconstrained PMF runs, with five sources identified: HOA, COA, solid fuel OA (SFOA), SVOOA and LVOOA.

Comment 6 from Referee.

Line 317 – I don't see BBOA, HOA or COA in Figure 4a.

author's response

Figure 4.a shows only SVOOA and LVOOA values. The caption on the plot has been modified. The clouds of dots show the f44:f43 and f44:f60 values for the OA measurements (the adding of these clouds was required for another referee).

Comment 7 from Referee. Lines 318-319 - How do the connecting lines display variability? Are they intended to make the differences between the cluster of LVOOA and the wider spread of SVOOA points more obvious?

author's response

Yes, this is has been explained with more detail in the manuscript.

author's changes in manuscript

Figure 4.a shows that LVOOA, while having different values between solutions, is found in distinct areas of the plot (connecting lines are used to make the SVOOA variability clearer), whereas SVOOA shows values of f44 vs f43 with high variability.

Comment 8 from Referee.

Lines 328-329 – could cloud processing also contribute to the higher 44 and lower 60?

author's response

Yes, paragraph has been edited in the manuscript.

author's changes in manuscript

BBOA evolution has been frequently observed with high f44 and low f60 values due to due to aging, oxidation and cloud processing (Huffman et al., 2009;Cubison et al., 2011). Thus, it was possible to obtain a variety of BBOA for the different seasons of the year, ranging from a fresh BBOA with a high f60 during autumn to a more oxidised BBOA with a low f60 during summer.

Comment 11 from Referee.

Lines 359 – 363 – you use "possible to observe" in a couple of places in this paragraph. I suggest re-writing those segments.

author's response

The paragraph has been edited on the manuscript.

author's changes in manuscript

Figure 5 shows the WD/WE ratios, where it is possible to observe NOx/$\Delta$CO ratios of 1.25, 1.35 and 1.136 for March-Dec, summer and autumn, respectively, suggesting diesel with a higher contribution during WD compared to petrol. These findings are confirmed by the CO/HOA ratios, which for the same periods of time, are lower than one (0.8, 0.45 and 0.9) suggesting a lower contribution of petrol during weekdays compared

to diesel. In spring, there are no considerable changes to the WD/WE ratios, although a higher contribution of petrol is shown during WD with values of 1.28 for CO/HOA and low diesel contribution.

Comment 13 from Referee.

Line 375 –How do you derive PM1? It is a large unstated assumption if you mean that the ACSM measures PM1. It will measure a larger fraction of the PM1, but it is not a PM1 measurement. author's response

PM1 composition is mainly conformed by OA, SO4, NO3, NH4 and BC, these components were mentioned on the manuscript. Also, Fig. 6 has been updated.

author's changes in manuscript

Considering that PM1 is composed mainly of OA, SO4, NO3, NH4 and BC, it is possible to analyse the PM1 composition during PM2.5 high concentrations (Fig. 6.b).

Comment 14 from Referee.

Line 379 – Another unsubstantiated claim. Explain how you come to the conclusion that secondary aerosols are the main contributors to PM2.5 from particles smaller than the upper limit of the ACSM. Which components of Figure 6 do you consider secondary and which do you consider primary? author's response

BBOA, HOA COA and BC are considered primary while SVOOA, LVOOA, NO3, NH4 and SO4 are considered secondary. The text and figure in the manuscript have been modified.

author's changes in manuscript

Defining BBOA, HOA COA and BC as primary and SVOOA, LVOOA, NO3, NH4 and SO4 as secondary aerosols, the main PM1 contributors to PM2.5 concentrations are secondary aerosols with a total contribution of 61% (Fig. 6.c).

All the minor comments such as typos were accepted and modified.

[Figure]

[Figure]

Figure 4: f44 vs f43 (a) and f44 vs f60 (b) plots for different periods of time.

**Fig. 1.** Figure 4

[Figure]

Figure 6: Daily PM2.5 concentrations (a) daily (b) and total PM1 composition (c), purple line in Fig. 6.c separates secondary and primary aerosols.

**Fig. 2.** Figure 6